# An on-chip phased array for non-classical light

Volkan Gurses ●[1,2] ✉, Samantha I. Davis ●[2,3], Raju Valivarthi[2,3], Neil Sinclair[3,4], Maria Spiropulu ●[2,3] & Ali Hajimiri ●[1]

Quantum science and technology can offer fundamental enhancements in sensing, communications and computing. The expansion from wired to wireless links is an exciting prospect for quantum technologies. For classical technologies, the advent of phased arrays enabled directional and adaptive wireless links by manipulating electromagnetic waves over free space. Here we demonstrate a phased array system on a chip that can receive, image and manipulate non-classical light over free space. We use an integrated photonic-electronic system with more than 1000 functional components on-chip to detect squeezed light. By integrating an array of 32 sub-wavelength engineered metamaterial antennas, we demonstrate a direct free-space-to-chip interface for reconfigurable quantum links. On the same chip, we implement a large-scale array of quantum-limited coherent receivers that can resolve non-classical signals simultaneously across 32 channels. With coherent readout and manipulation of these signals, we demonstrate 32-pixel imaging and spatially configurable reception of squeezed light over free space. Our work advances wireless quantum technologies that could enable practical applications in quantum communications and sensing.

The science and engineering of quantum systems have expanded in the last two decades to realize technologies that can manipulate quantum information[1–3]. Miniaturization and scaling of quantum systems with on-chip integration are crucial to accelerate their use cases toward practical applications[4,5]. Free-space-interfaced integrated systems enable wireless technologies including free-space sensors[6,7], imagers[8,9], and communication transceivers[10,11]. The expansion from wired to wireless links for classical information technologies led to numerous advancements from mobile devices[12] to the Internet-of-Things[13] and facilitated the proliferation of information technologies[14]. For a similar transformation to happen for quantum information technologies, the networking of integrated quantum systems needs to move beyond wired links[15–18].

In most wireless links, phased arrays are used to enable spatio-temporally configurable signal reception or transmission with high signal gain[19,20]. A phased array is a coherent array of antenna elements capable of transmitting or receiving electromagnetic waves. Through the interference of the electromagnetic waves transmitted or received by each element, arbitrary wavefronts can be engineered[19,20]. The first phased arrays were implemented to control electromagnetic waves at radio frequencies[21], facilitating RADAR[7], wireless communications[11], remote sensing[22] and radio astronomy[23]. In the last decade, advancements in nanophotonics enabled large-scale phased arrays at optical frequencies[24], opening up applications including LiDAR[25,26], 3D imaging[8], and free-space optical communications[10].

All previous developments in reconfigurable antenna arrays have been achieved only with classical states of light. Extending phased arrays to the non-classical domain[27,28] could enable potentially interesting use cases for quantum information technologies, such as wirelessly-interfaced quantum systems (Fig. 1a) and reconfigurable wireless quantum links (Fig. 1b). However, due to the high coupling loss and noise in conventional transceivers, there has not been a phased array system capable of interfacing with non-classical light.

[1]Division of Engineering and Applied Science, California Institute of Technology, Pasadena, CA, USA. [2]Division of Physics, Mathematics and Astronomy, California Institute of Technology, Pasadena, CA, USA. [3]Alliance for Quantum Technologies (AQT), California Institute of Technology, Pasadena, CA, USA. [4]John A. Paulson School of Engineering and Applied Sciences, Harvard University, Cambridge, MA, USA. ✉e-mail: gurses@caltech.edu

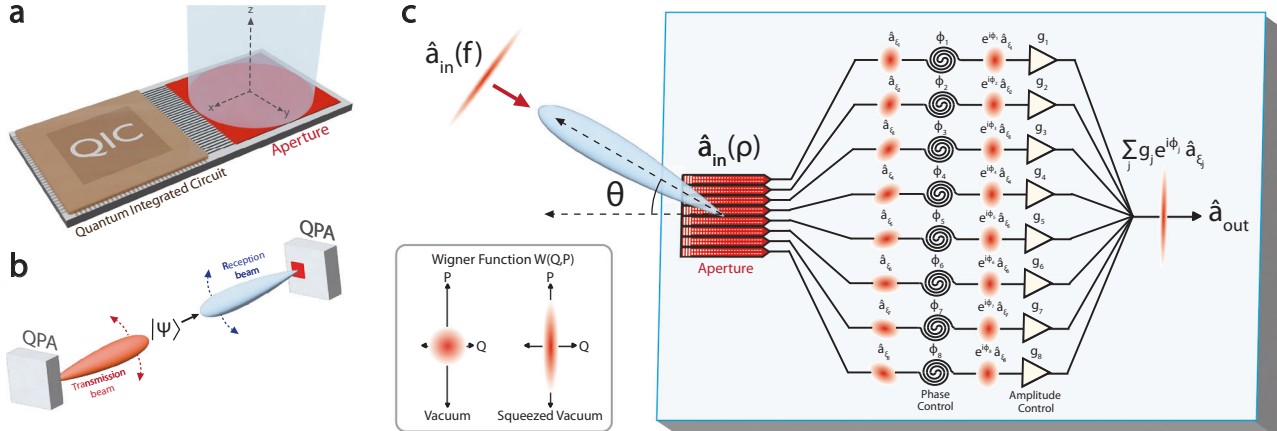

**Fig. 1 | Quantum phased arrays. a** Conceptual illustration of a wirelessly-interfaced quantum integrated circuit. **b** Conceptual illustration of a wireless quantum link with phased arrays. A phased array transmitter transmits a quantum state |Ψ⟩ to a phased array receiver over free space. **c** Conceptual illustration of beamforming on squeezed light with an eight-element phased array receiver. An input field ($\hat{a}_{\text{in}}(f)$) in a squeezed state is transmitted to a phased array receiver over free space. The field incident to the aperture ($\hat{a}_{\text{in}}(\rho)$) is spread out over the aperture with a uniform phasefront, resulting in high geometric loss per pixel mode. After applying a phase ($\phi_j$) and amplitude weight ($g_j$) to each pixel mode ($\hat{a}_{\mathcal{E}_j}$), the pixel modes are combined to recover the original squeezed state. Squeezed states are represented by their Wigner functions in phase space, where $Q$ and $P$ represent the field quadratures (see inset).

In this work, we realize a phased array system-on-chip with quantum-limited performance that can receive and control non-classical light. The 32-channel silicon photonic-electronic system provides a low-loss, low-noise and scalable free-space-to-chip interface for non-classical light. Simultaneous readout of the signals across all 32 channels enables 32-pixel imaging of squeezed light transmitted over free space. Coherent manipulation of these signals allows us to establish reconfigurable free-space links for squeezed light.

## Results

### Quantum phased arrays

A quantum phased array is a quantum coherent array of antenna elements capable of transmitting or receiving quantum fields, such as non-classical light. A phased array receiving non-classical light is illustrated in Fig. 1c. A quantized electromagnetic field is transmitted over free space to a phased array receiver with $N$ elements. The field incident to the aperture is represented by the local bosonic operator $\hat{a}_{\text{in}}(\rho)$, where $\rho$ represents the spatial aperture coordinates. Due to diffraction over free space, the incident field is spread out over the aperture and a portion of the field is coupled onto each antenna element. The antenna elements define a set of $N$ pixel modes, $\{\mathcal{E}_j(\rho)\}$, each with an associated bosonic operator $\hat{a}_{\mathcal{E}_j} = \int \mathcal{E}_j^*(\rho)\hat{a}_{\text{in}}(\rho)d\rho$. The pixel modes are combined after applying a phase, $\phi_j$, and amplitude weight, $g_j$, to each mode. The combined output field is described by,

$$\hat{a}_{\text{out}} = \sum_j g_j e^{i\phi_j}\hat{a}_{\mathcal{E}_j} = \int \mathcal{A}^*(\rho)\hat{a}_{\text{in}}(\rho)d\rho, \tag{1}$$

where $\mathcal{A}(\rho) = \sum_j g_j e^{-i\phi_j}\mathcal{E}_j(\rho)$. The set of applied phases, amplitude weights, and pixel modes gives rise to a reconfigurable array mode function $\mathcal{A}(\rho)$ that can be used to engineer the wavefunction of the incident field (see theory in Methods)[29,30].

The output field can be expressed in terms of the angle $\theta$ from normal incidence to the aperture,

$$\hat{a}_{\text{out}} \propto \int \text{SF}(f)\hat{a}_{\text{in}}(f)df, \tag{2}$$

where $f = \sin\theta/\lambda$, $\text{SF}(f) = \int \exp(-i2\pi\rho \cdot f)\mathcal{A}(\rho)d\rho$, and $\hat{a}_{\text{in}}(f)$ represents the input field in the far-field limit. In Eq. (2), SF($f$) corresponds to the space factor, or array factor for discrete antenna elements, in

classical antenna theory[19,20]. Beamforming refers to the calibration of the phase settings to form a main lobe, or beam, in the radiation pattern at a desired angle[31]. After beamforming, the beam can be steered to a different angle by applying a progressive phase shift to the elements[19,20] (see theory in Methods).

### Photonic-electronic system

We realize a proof-of-concept phased array system-on-chip that operates at the quantum-limited sensitivity and interfaces with non-classical light to demonstrate different functionalities of our concept. The system is implemented using a commercial silicon photonics process packaged with silicon electronics, as shown in Fig. 2. The system is designed to realize as many functions on chip as possible, with more than 1000 functional components integrated on a 3 mm × 1.8 mm footprint.

Compared to conventional phased arrays, our phased array system needs to be designed to introduce as little loss and noise as possible to minimize quantum decoherence. The most significant loss for free-space-interfaced systems is geometric loss due to the mode mismatch between an incident beam and the receiving aperture[19,32]. In the case of a collimated beam, the mode mismatch is caused by beam divergence. To mitigate this, the aperture needs to be large enough to be able to match the amplitude and phase profile of the incident beam. Phased arrays enable arbitrarily large apertures that can be mode matched by arraying multiple antennas and coherently combining the received signals[19,20]. Our system demonstrates this for squeezed light by arraying 32 nanophotonic antennas. To minimize the loss, the aperture also needs to be fully filled without any gaps in the active area. We achieve this by demonstrating a metamaterial antenna (MMA) design that acts as a rectangular building block and can be sized to fill the aperture (see chip design in Methods).

An array of 32 MMAs fill an aperture with a 550 × 550 μm² footprint and more than 500,000 sub-wavelength-engineered grating elements, each of which scatter light to interface with free space. This active area is large enough for low-loss free-space coupling to the chip over meter-scale distances with off-the-shelf fiber collimators (see Sections IA and IXA in the Supplementary Information).

The simulated 3D radiation pattern for the MMA design is shown in Fig. 3a. The aperture can be characterized by its geometric loss given an incident beam and its insertion loss, which includes the propagation loss in the MMA and loss due to downward scattering. The MMA has a

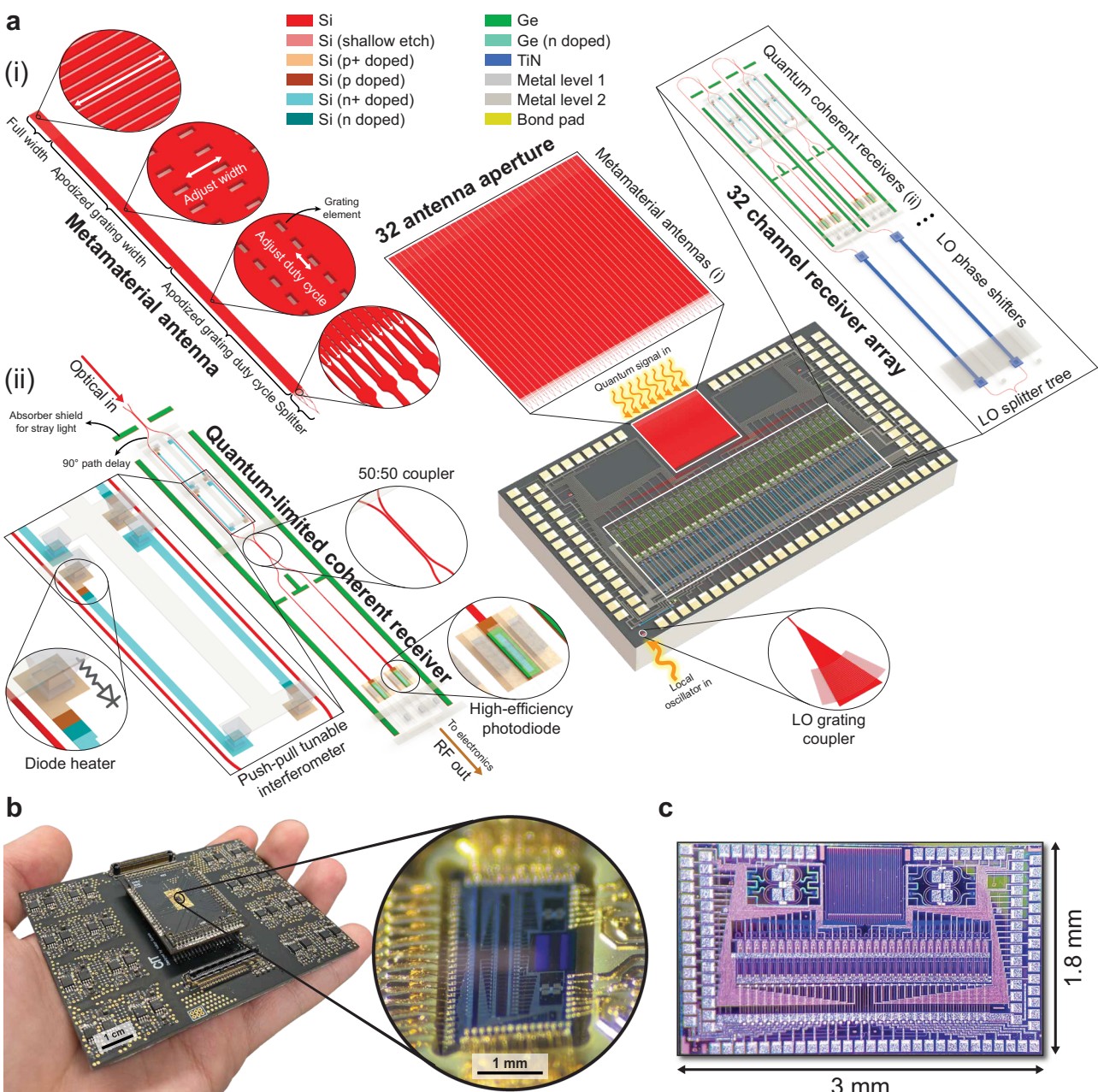

**Fig. 2 | Photonic-electronic system. a** Diagram of the photonic integrated circuit (PIC) illustrating the key building blocks, including i) the metamaterial antenna (MMA) and ii) the quantum(-limited) coherent receiver (QRX). An array of 32 MMAs couple non-classical light from free space to on-chip waveguides, followed by an array of 32 QRXs that measure the light via homodyne detection. An array of 32 thermo-optic phase shifters (TOPS) applies a phase shift to the local oscillator at each QRX. **b** Image of our PIC packaged with co-designed electronics, demonstrating the compact form factor of the system. The PIC is wirebonded to an interposer, which is plugged into a radio-frequency motherboard that hosts a 32-channel TIA array and the CMRR auto-correction circuit. **c** Die photo of the PIC showing a footprint of 3 mm × 1.8 mm.

measured (simulated) insertion loss of 3.82 dB (3.78 dB). The 32-antenna aperture mode matched to a collimated beam with a beam diameter of 200 µm, which is the beam diameter used in the experiments, has a measured (simulated) geometric loss of 1.14 dB (1.35 dB). This is at least an order of magnitude lower than those of the previously reported on-chip aperture designs[33,34], which affords interfacing free-space quantum optics with photonic integrated circuits (PICs) (see Section IA in the Supplementary Information).

The waveguides after the antennas are path-length matched and are connected to 32 quantum(-limited) coherent receivers (QRXs). Each QRX comprises a push-pull tunable Mach-Zehnder interferometer (MZI), a pair of balanced Ge photodiodes, a transimpedance

amplifier (TIA), and a common-mode rejection ratio (CMRR) auto-correction circuit, whose output is fed back to the MZIs to automatically correct the imperfect CMRR of each QRX caused by the fabrication variations on the PIC (see chip design in Methods). The MZI interferes a signal field with a strong local oscillator (LO) for homodyne detection. The LO is coupled to the chip with a grating coupler and is split into 32 channels with a 1-to-32 splitter tree. The LO input to each channel hosts a thermo-optic phase shifter (TOPS) for phase tuning. Each output of the MZI is sent to a photodiode, and the currents at the outputs of the photodiodes are subtracted and amplified by the TIA.

The performance of a QRX is quantified by its insertion loss, common-mode rejection ratio (CMRR), shot noise clearance (SNC), LO

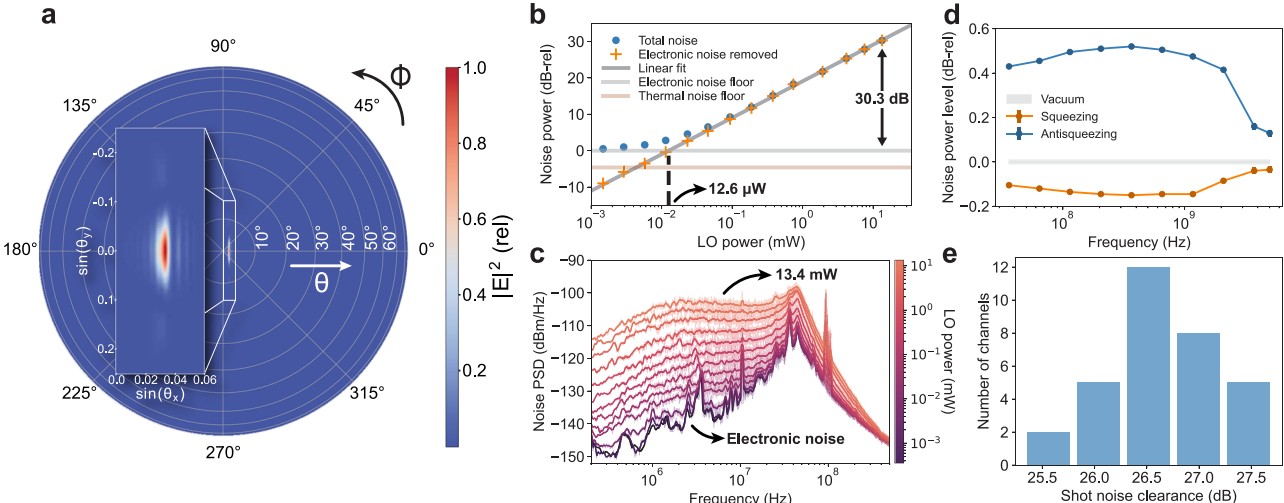

**Fig. 3 | System characterization. a** Simulated far-field radiation pattern of the antenna. The radiation pattern has no grating lobes, namely scattering to higher diffraction orders, showing that the MMA is sub-wavelength engineered for diffraction-limited performance. **b** Noise powers of a single-channel QRX in the 32-channel system integrated over its 3-dB bandwidth for different LO powers, characterizing the shot noise clearance and LO power knee. A linear regression fit is applied to the data above the LO power knee to obtain a near-unity gradient of 1.004 ± 0.006, showing that the QRX noise floor is limited by the signal shot noise. **c** Output noise spectra of a single QRX for different LO powers ranging from 0 to 13.4 mW, characterizing the shot-noise-limited bandwidth. **d** Squeezed light detection with a single QRX using a high-speed TIA, showing squeezing and anti-squeezing measured up to 5 GHz with a shot-noise-limited bandwidth of 3.70 GHz. **e** Shot noise clearance distribution across all channels measured with 1.54 mW LO power at each channel.

power knee ($P_{knee}$), 3-dB bandwidth ($BW_{3dB}$) and shot-noise-limited bandwidth ($BW_{shot}$)[15,35–37]. We first measure a single channel of the 32-channel QRX array used in the experiments to characterize these specifications, as shown in Fig. 3b, c. The measured insertion loss is 1.58 dB, limited by the quantum efficiency of the photodiodes. A time-averaged CMRR over 10 seconds is measured to be 90.2 dB at 1.1 MHz. The measured SNC is 30.3 dB, $P_{knee}$ is 12.6 μW, $BW_{3dB}$ is 10.4 MHz and $BW_{shot}$ is 381 MHz (see Section IB in the Supplementary Information).

While this gives one of the highest SNCs demonstrated with an integrated coherent receiver in the literature[15,35], the measured bandwidth is relatively low due to the intrinsic trade-off between noise floor and bandwidth in the TIA design. To show the high-speed detection capability of our PIC with on-chip photodiodes, a single channel of the PIC is packaged with a bare die TIA, and the same characterization sweeps are performed (see Section IB5 in the Supplementary Information)[38]. In this high-bandwidth configuration, squeezed vacuum is injected as signal, and noise power fluctuations below and above the shot noise floor are measured up to 5 GHz with a $BW_{shot}$ of 3.70 GHz, as shown in Fig. 3d.

Furthermore, we characterize all 32 channels while they work simultaneously. With 1.54 mW of LO power, the SNCs across all of the channels are measured, and the SNC distribution is plotted in a histogram as seen in Fig. 3e. The SNC variation is low, with a median SNC of 26.6 dB, a minimum SNC of 25.3 dB, and a maximum SNC of 27.7 dB. All channels operate well into the shot noise limited regime, showing that the quantum-limited performance of the QRX is achieved at scale (see Section II in the Supplementary Information).

**Squeezed light imaging**
We first operate the system as a 32-pixel quantum-limited coherent imager. Broadband squeezed vacuum is generated off-chip using a fiber-coupled periodically poled lithium niobate (PPLN) waveguide at a central wavelength of 1550 nm, as shown in Fig. 4a (see squeezed light generation in Methods). The squeezed light is sent to a fiber collimator with a 200 μm beam diameter and is transmitted to the chip over free space. At the chip aperture, the squeezed light is spatially distributed across the 32 antennas with a Gaussian amplitude profile, $u_0(\rho)$. A portion of the squeezed light is coupled into each MMA, which has the

associated pixel mode $\mathcal{E}_j(\rho)$ and bosonic operator $\hat{a}_{\mathcal{E}_j}$, where $j \in \{1, \cdots, 32\}$. The pixel mode is interfered with an LO in a QRX, which outputs a voltage proportional to the phase-dependent quadrature of the pixel mode,

$$\hat{Q}_j(\phi_j) = \frac{1}{2}\left(\hat{a}_{\mathcal{E}_j}e^{-i\phi_j} + \hat{a}_{\mathcal{E}_j}^\dagger e^{i\phi_j}\right), \tag{3}$$

where $\phi_j$ is the phase of the $j$th pixel mode relative to the LO. For an input squeezed vacuum state, the quadrature mean is $\langle \hat{Q}_j(\phi_j)\rangle = 0$, and the quadrature variance is,

$$\text{Var}\left(\hat{Q}_j(\phi_j)\right) = \frac{\eta_j}{4}\left(e^{-2r}\cos^2\phi_j + e^{2r}\sin^2\phi_j\right) + \frac{1-\eta_j}{4}, \tag{4}$$

where $r$ is the squeezing parameter and $\eta_j$ is the effective efficiency of channel $j$, which includes the effects of source loss, free-space loss dominated by geometric loss, on-chip loss, and radio-frequency (RF) loss. Here, $\phi_j = 0$ and $\phi_j = \pi/2$ correspond to the squeezed and the antisqueezed quadratures, respectively.

To image the squeezed light, the output voltages are read out to a 32-channel digitizer (see data acquisition in Methods). A 0.5 Hz phase ramp is applied to the LO off-chip to acquire voltage samples over various phases, and sample means and variances are calculated over sets of 260,000 voltage samples. The time evolution of the sample means and variances for all 32 pixel modes are shown in Fig. 4c. Without phase locking, thermal drifts in the fiber-optic setup give rise to nonuniform phase fluctuations on top of the phase ramp, which are coherent across all channels. The corresponding Wigner functions for the source and the pixel modes are shown in Fig. 4b, d, respectively (see Wigner function calculation in Methods).

**Free-space quantum links**
Next, we operate the system as a reconfigurable quantum receiver. A QRX is used to apply a phase shift and amplitude weight to the quadrature of each pixel mode by varying the phase of the LO and the amplitude of the RF output. After coherently combining the QRX outputs, the combined RF output is a voltage proportional to the

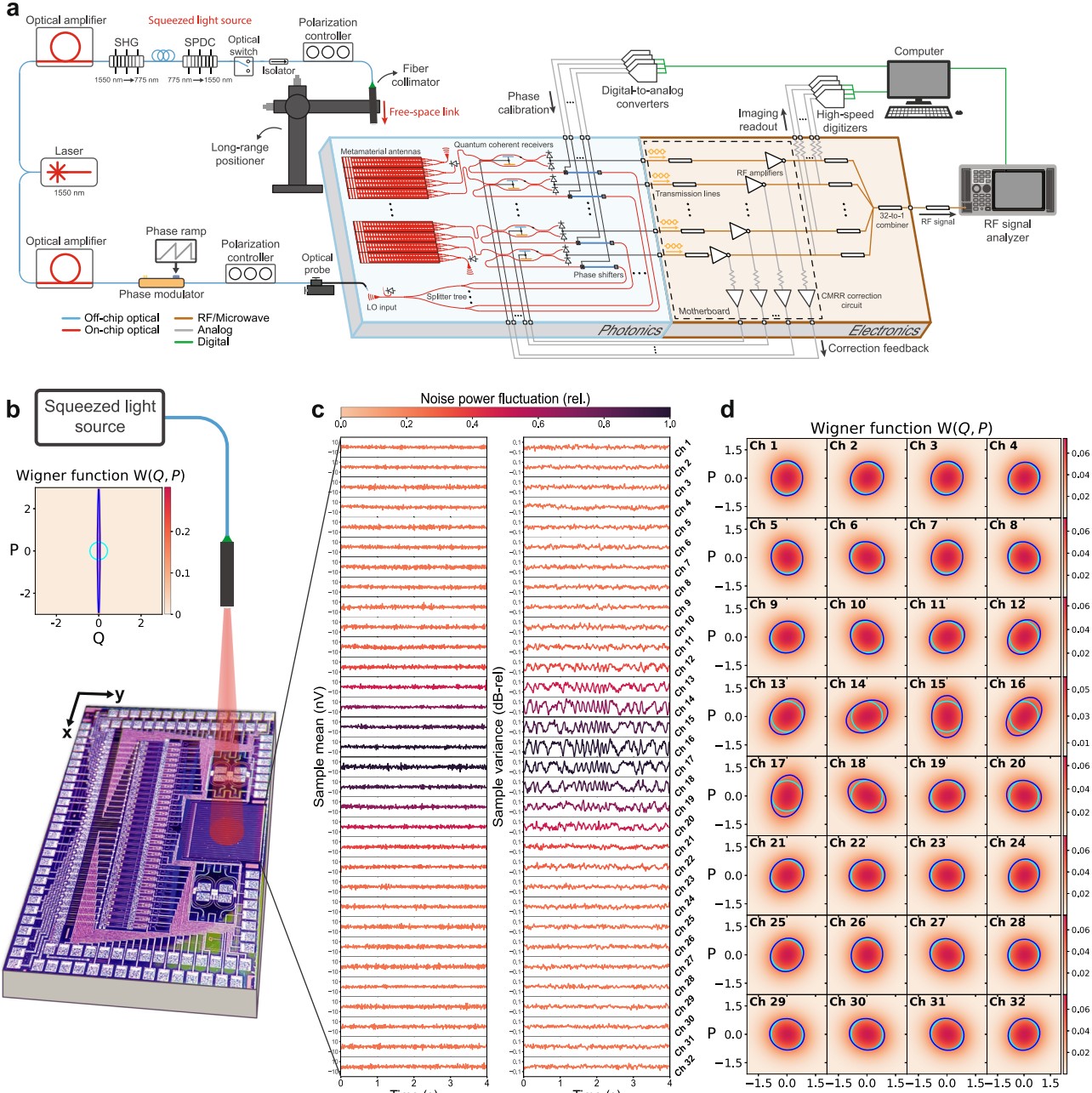

**Fig. 4 | Squeezed light imaging. a** Experimental setup for the squeezed light measurements. Squeezed light is generated off-chip and transmitted over free space to the chip (blue, Photonics), which is interfaced with electronics (orange, Electronics) for processing. **b** Illustration of squeezed light transmitted to the chip, showing the Wigner function of the generated squeezed vacuum state as a function of the quadrature observables $(Q, P)$ and the experimental squeezing parameter

$(r = 1.95)$. **c** Sample means and variances of the channel output voltages as a function of time. For each channel, the sample variances are normalized to the mean variance. **d** Wigner functions of the 32 pixel modes characterized simultaneously as a function of the squeezing parameter $(r = 1.95)$, phase, and geometric efficiency for each channel. The dark and light blue contours correspond to the half-maximum points of the squeezed vacuum and vacuum states, respectively.

quadrature of the engineered output field, $\hat{a}_{out}$ in Eqs. (1) and (2) (see implementation in Methods),

$$\hat{Q}_{out} = \sum_j g_j \hat{Q}_j(\phi_j) = \frac{1}{2}\left(\hat{a}_{out} + \hat{a}_{out}^\dagger\right). \qquad (5)$$

Beamforming on squeezed light with the system is illustrated in Fig. 5a. After calibrating the LO phases for all 32 channels, squeezed light with $r = 0.76$ (6.61 dB generated squeezing) is transmitted to the chip through the fiber collimator. A 1 Hz phase ramp is applied to the LO before coupling to the chip, and the outputs of the channels are

coherently combined with a 32:1 RF power combiner. The combined output signal is sent to an RF signal analyzer, which measures the noise power proportional to the variance of Eq. (5) (see beamforming and data acquisition in Methods).

Noise powers for squeezed vacuum and vacuum states are measured for various numbers of combined channels. Disconnected channels have zero amplitude weight and connected channels have approximately uniform amplitude weights. For a given set of amplitude weights, the phase calibration step in beamforming maximizes the modal overlap of the array mode function $\mathcal{A}(\rho)$ and the squeezed input mode $u_0(\rho)$, corresponding to the geometric efficiency

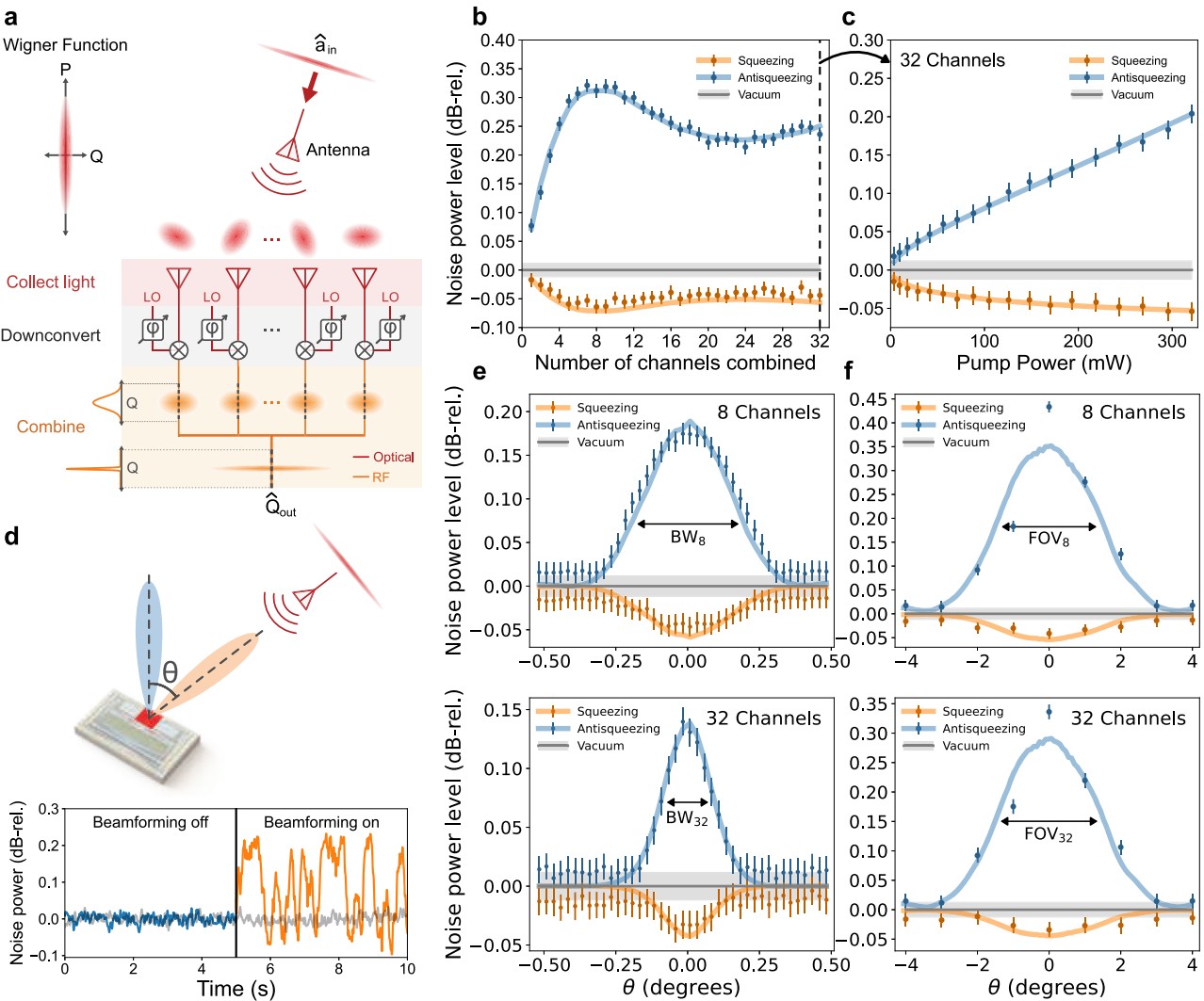

**Fig. 5 | Free-space quantum links. a** Conceptual illustration of beamforming on squeezed light with the chip, where $\hat{a}_{in}$ represents the input field and $\hat{Q}_{out}$ is the quadrature proportional to the combined output signal at RF. **b** Squeezing and antisqueezing levels as a function of the number of combined channels relative to the vacuum level after the chip is beamformed toward the squeezed light transmitter. **c** Squeezed light source characterization showing squeezing and antisqueezing levels as a function of source pump power for 32 combined channels. **d** Demonstration of reconfigurable free-space quantum links, illustrating the lack of squeezed light signal when the receiver is beamformed toward empty space (blue) and the reception of the signal when the receiver is beamformed toward the transmitter (orange). The gray trace is the vacuum signal. **e** Squeezing and antisqueezing levels characterizing the beamwidth of the link for 8 and 32 combined channels. **f** Squeezing and antisqueezing levels characterizing the field of view of the receiver for 8 and 32 combined channels. In (**b**, **c**, **e**, **f**), the orange and blue solid lines are fits of the data to a model obtained from the classical characterization of the corresponding measurement.

$\eta_g = |\int \mathcal{A}^*(\rho) u_0(\rho) d\rho|^2$. Therefore, by configuring the amplitude weights in the array mode function, the number of combined channels sets the geometric efficiency. While in the experiment, we had binary control over the amplitude weights, amplitude control can be made more precise by tuning the SNC of each channel either by controlling the LO amplitude or controlling the electronic noise floor. The squeezing and antisqueezing levels relative to the shot noise level for each channel combination are shown in Fig. 5b. The squeezing (antisqueezing) improves from −0.017 (+0.077) ±0.012 dB-rel for a single channel to −0.064 (+0.312) ±0.012 dB-rel for eight combined channels, corresponding to an increase in the geometric efficiency by a factor of 4.5. For more than eight combined channels, the observed squeezing and antisqueezing decreases due to reduced modal overlap. The squeezing and antisqueezing increase again for more than 24 combined channels, most likely due to parasitic effects at RF (see channel combination and measurement characterization Methods).

After beamforming on 32 channels, we perform a source characterization to confirm that the combined RF signals correspond to the measurement of the squeezed light source. For all 32 channels combined, the squeezing and antisqueezing levels for various source pump powers ($P$) are shown in Fig. 5c. The solid lines correspond to a least-squares fit of the data to a model, where the effective efficiency of the system, $\eta$, and the spontaneous parametric downconversion (SPDC) efficiency, $\mu = r/\sqrt{P}$, are taken as floating parameters. We obtain $\eta = 0.016$ and $\mu = 0.038$ [mW]$^{-1/2}$, which is consistent with the independently characterized SPDC efficiency of the source (see theoretical modeling in Methods).

Establishing directional free-space links for squeezed light with our system is illustrated in Fig. 5d. Squeezed light is transmitted to the system over free space. First, the phased array forms a reception beam directed away from the transmitter and no light is detected (blue). The phase settings are then reconfigured by applying a linear phase mask to the LO phase shifters, which steers the reception beam toward the transmitter and forms a free-space link (orange). In the first five seconds, no squeezed light signal is observed, demonstrating successful spatial filtering. In the next five seconds, after the reception beam is

electronically steered toward the transmitter, the noise power modulations of the squeezed light are observed.

The spatial selectivity, namely beamwidth, of a link is characterized in Fig. 5e. Phase calibration is performed for squeezed light with $r = 0.61$ (5.31 dB generated squeezing) transmitted to the chip at a normal angle of incidence. The angle of incidence ($\theta$) of the squeezed light is swept while the chip is kept beamformed to normal incidence ($\theta = 0$). The squeezing and antisqueezing levels as a function of $\theta$ are shown in Fig. 5e for 8 and 32 combined channels. The beamwidths (BW$_n$) corresponding to 50% efficiency (3 dB loss) are $0.41 \pm 0.02$ degrees and $0.20 \pm 0.02$ degrees with $n = 8$ channels and $n = 32$ combined channels, respectively. The beamwidth decreases with the number of combined channels, demonstrating the expected increase in spatial selectivity as the array is scaled up[19,20] (see measurement characterization in Methods).

The reconfigurability of the links over the field of view (FoV) of the system is demonstrated in Fig. 5f. At each of the nine different angles of incidence ($\theta$), phase calibration is performed and the optimal phase settings are recorded. Squeezed light with $r = 0.91$ (7.89 dB generated squeezing) is then transmitted to the chip at each angle, and a link is programmed by applying the LO phase shifter settings that form a reception beam at the corresponding angle. The squeezing and antisqueezing levels for each angle are shown in Fig. 5f for 8 and 32 combined channels. The FoV for squeezed light corresponding to 50% efficiency (3 dB loss) is $2.32 \pm 0.12$ degrees and $2.66 \pm 0.25$ degrees with 8 channels and 32 combined channels, respectively. The FoV stays the same as the array is scaled up, matching the single antenna radiation pattern[19,20] (see measurement characterization in Methods).

## Discussion

We have demonstrated an on-chip phased array with quantum-limited performance for receiving and manipulating non-classical light in a large-scale silicon photonic-electronic system. With our system, we have demonstrated, for the first time, 32-pixel imaging of squeezed light and reconfigurable free-space-to-chip links for squeezed light. These functionalities are enabled by a fully-filled and low-loss metamaterial aperture and the first large-scale coherent receiver array capable of resolving non-classical signals, with up to 30.3 dB SNC, 90.2 dB CMRR and 3.70 GHz bandwidth.

In our work, the amount of measured squeezing was limited by ~15 dB system loss, dominated by ~8 dB off-chip source loss. Based on the improvements in component losses already demonstrated in the literature[39,40], there is a clear path toward sub-dB (down to 0.626 dB) loss for our system (see Section VIII in the Supplementary Information). Thanks to the modular architecture of our system, the aperture can be scaled by duplicating individual channels to achieve longer-distance free-space links. Additionally, phase locking can be implemented[41], or the signal and LO can be transmitted over the same free-space channel, to remove phase fluctuations.

Together with a transmitter counterpart to our system, our approach could enable wireless quantum technologies based on phased array transceivers. Quantum-enhanced sensors could be built with phased arrays by placing a highly transmissive or reflective sample in a free-space link and imaging the sample with non-classical light[42,43]. With the current chip, up to 18% improvement below the shot noise floor in sensitivity is possible. With 0.626 dB system loss, the sensitivity improvement could reach 87%, corresponding to 8.75 dB measured squeezing, potentially enabling practical quantum enhancement in imaging[42,43], LiDAR[44,45] and microscopy[46].

Wireless quantum communication networks[47] could also be built with phased array transceivers forming the nodes of a network. With our current chip, continuous-variable quantum key distribution[48] is possible with up to 3.81 Mbps secure key rate and a non-zero key rate distance of 56.3 cm. This distance could be further extended to 1.69 km and 39.2 km, limited by the atmospheric attenuation, with

reticle-scale ($30 \times 30$ mm$^2$) and wafer-scale ($300 \times 300$ mm$^2$) apertures, respectively, putting long-distance quantum communications with chip-scale devices within reach (see Section IX in the Supplementary Information).

## Methods
### Theory

Consider a quantized electromagnetic field $\hat{E}$ transmitted over free space to a phased array receiver. The field can be decomposed into positive and negative frequency components, $\hat{E} = \hat{E}^+ + \hat{E}^-$, where $\hat{E}^+$ and $\hat{E}^-$ are Hermitian conjugates. The positive frequency component of the field at the aperture can be expressed as,

$$\hat{E}^+(\rho, t) = \sqrt{\frac{\hbar\omega}{2\epsilon_0 V}} \sum_n u_n(\rho) e^{-i\omega t} \hat{a}_{u_n}, \tag{6}$$

where $\rho = (x, y)$ are the transverse spatial coordinates in the aperture plane (see Fig. 1a), $\omega$ is the frequency, and $V$ is the quantization volume[28,49]. The field is expanded over a complete set of orthonormal modes $\{u_n(\rho)\}$, such as Hermite-Gaussian modes, that correspond to photon-wavefunctions in second quantization[50]. Each mode has an associated pair of bosonic operators $\hat{a}_{u_n}$ and $\hat{a}_{u_n}^\dagger$ satisfying $[\hat{a}_{u_n}, \hat{a}_{u_m}^\dagger] = \delta_{n,m}$. In Eq. (6), we assume a monochromatic treatment of the field. We note that the squeezed light generated in the experiments is broadband and that our analysis can be extended to multiple spectral modes.

The field incident to the aperture can be represented by the local bosonic operator, $\hat{a}_{\text{in}}(\rho) = \sum_n u_n(\rho) \hat{a}_{u_n}$. The aperture is divided into $N$ antenna elements, which define a set of $N$ pixel modes $\{\mathcal{E}_j(\rho)\}$ each with an associated pair of bosonic operators $\hat{a}_{\mathcal{E}_j}$ and $\hat{a}_{\mathcal{E}_j}^\dagger$. The pixel modes are combined after applying a phase shift, $\phi_j$, and an amplitude weight, $g_j$, to each pixel mode, where $\sum g_j^2 = 1$. The combined output field is described by Eq. (1),

$$\hat{a}_{\text{out}} = \sum_j g_j e^{i\phi_j} \hat{a}_{\mathcal{E}_j} = \int \mathcal{A}^*(\rho) \hat{a}_{\text{in}}(\rho) d\rho,$$

where $\hat{a}_{\mathcal{E}_j} = \int \mathcal{E}_j^*(\rho) \hat{a}_{\text{in}}(\rho) d\rho$ is substituted to obtain the array mode function, $\mathcal{A}(\rho)$,

$$\mathcal{A}(\rho) = \sum_j g_j e^{-i\phi_j} \mathcal{E}_j(\rho) \approx g(\rho) e^{-i\phi(\rho)}. \tag{7}$$

The approximation in Eq. (7) is taken in the small pixel limit. In this limit, the pixel modes approach $\mathcal{E}_j(\rho) \approx \delta(\rho - \rho_j)$, where $\rho_j$ represents the coordinates at the center of the $j$th pixel, the set of amplitude weights and phase shifts approach continuous phase, $\phi(\rho_j)$, and amplitude, $g(\rho_j)$, distributions, and the sum becomes an integral over $\rho_j$. The reconfigurable array mode function can be used to engineer the state of the incident field to enable functionalities such as mode matching, beamforming and beamsteering, and more generally quantum state engineering[29,30].

**Mode matching and discrimination.** Consider a quantum state transmitted over free space to a phased array receiver in mode $u_n(\rho)$, with all other modes in the vacuum state. In our experiments, we transmit a squeezed vacuum state in the Gaussian mode $u_0(\rho)$. Due to diffraction, the incident field is spread out across the aperture, with multiple vacuum modes $u_{m \neq n}(\rho)$ coupling into each antenna element. This results in a low geometric efficiency for each pixel mode, $\eta_{g,j} = |\int \mathcal{E}_j^*(\rho) u_n(\rho) d\rho|^2$. The array can mode match to $u_n(\rho)$ by configuring the array mode function, $\mathcal{A}(\rho)$, through weighted combination of the pixels such that the vacuum modes destructively interfere. The geometric efficiency of the array is described by $\eta_g = |\int \mathcal{A}^*(\rho) u_n(\rho) d\rho|^2$. Due to the orthonormality of the input basis,

unity geometric efficiency can be achieved by setting $\mathcal{A}(\rho) = u_n(\rho)$, resulting in $\hat{a}_{out} = \hat{a}_n$. For multimode fields, the signal in a particular mode can be uniquely selected by setting $\mathcal{A}(\rho)$ to the desired mode function. In general, the reconfigurability of $\mathcal{A}(\rho)$ enables mode matching to quantum states in arbitrary modal profiles, such as multimode quantum states, to achieve unity geometric efficiency.

**Beamforming and beamsteering.** The field at the aperture can be expressed in terms of the input field in the far-field limit, $\hat{a}_{in}(f)$, using the generalized Huygen's principle[51], which can be approximated as,

$$\hat{a}_{in}(\rho) \propto \int e^{-i2\pi\rho f} \hat{a}_{in}(f) df, \qquad (8)$$

where $f = \sin\theta/\lambda$ is in terms of the angle from normal incidence ($\theta$) and the wavelength ($\lambda$). We assume a one dimensional linear array of antennas. The output field of the phased array in terms of the input field is given by Eq. (2),

$$\hat{a}_{out} \propto \int \mathrm{SF}(f)\hat{a}_{in}(f)df,$$

where $\mathrm{SF}(f) = \int e^{-i2\pi\rho f} \mathcal{A}^*(\rho)d\rho$ corresponds to the space factor of the array[19,20]. In antenna theory, the total far-field radiation pattern is formed by pattern multiplication of the space factor and the element factor[19]. For our system, the element factor corresponds to the radiation pattern of a single antenna.

For a discrete linear array with spacing $d$ and a progressive phase shift $\Delta\phi$ applied to the antenna elements, the radiation pattern features a primary lobe, or a beam, at $\theta_{max} = \arcsin[-(\lambda/2\pi d)\Delta\phi]$[19]. The phased array transmitter is the reciprocal counterpart to a phased array receiver, and for a phased array receiver (transmitter), the beam corresponds to the range of reception (transmission) angles over which the pixel modes constructively interfere for a given set of phases and amplitude weights applied to the pixels. Beamforming refers to the optimization of the phase and amplitude settings of a phased array receiver (transmitter) in order to form a reception (transmission) beam at a given angle. In the receiver case, beamforming is equivalent to maximizing the geometric efficiency of the array, $\eta_g$. By varying the progressive phase to shift $\theta_{max}$, the reception or transmission beam can be steered toward a different location[19,20].

**Implementation.** Quantum phased arrays could be implemented in various platforms for discrete-variable or continuous-variable quantum information. In integrated photonics, phase control in Eq. (2) can be implemented with phase shifters using, for instance, the thermo-optic effect. Amplitude control in weighted combination can be implemented with meshes of interferometers[52] or homodyne detection[53]. In the case of homodyne detection, the amplitude weights are applied by tuning the amplitude of the LOs or the gains of the electronic amplifiers. Each coherent receiver outputs an RF field that generates a voltage or current proportional to the phase-dependent quadrature of its pixel mode defined in Eq. (3). Leveraging the coherence of the receiver outputs across the array, the output signals are combined at RF, yielding a combined signal proportional to the quadrature $\hat{Q}_{out} = (\hat{a}_{out} + \hat{a}_{out}^\dagger)/2$ of an effective output field[49], $\hat{a}_{out}$, described by Eq. (1). For our system, since phase control is performed with the thermo-optic phase shifters on the LO side and amplitude control is performed with electronic gains, the reconfigurability of the system adds no loss to the received non-classical light. In our implementation, configuring $\mathcal{A}(\rho)$ is equivalent to shaping the spatial-mode profile of the LO throughout the array. Therefore, our system acts as a reconfigurable, coherent spatial filter, in addition to spectral filtering enabled by the choice of the LO spectral mode for each pixel.

## Chip fabrication and design

**Chip fabrication.** The PIC was fabricated with Advanced Micro Foundry using a 193 nm silicon-on-insulator (SOI) process. The process has two metal layers (2000-nm thick and 750-nm thick) for electronic routing, a titanium nitride heater layer, a 220-nm thick silicon layer, a 400-nm thick silicon nitride layer, germanium epitaxy, and various implantations for active devices. A process design kit (PDK) from the foundry was provided. The PIC was laid out using KLayout and Cadence Virtoso and was simulated using Lumerical for design verification.

**Chip design.** To interface with non-classical light, the aperture requires an antenna design that can be sized to fill the aperture without any gaps. This requires a design methodology that can increase the active area of the antenna arbitrarily in both lateral and longitudinal dimensions. Increasing the active area requires minimizing the scattering strength per unit area while abiding by the design rules to be foundry compatible with the silicon photonics process.

To increase longitudinal area, grating elements are placed around the antenna waveguide with apodized scattering strength. To increase lateral area, sixteen of these waveguide grating antennas are connected and parallelized. The 0.82 μm wide waveguides keep a single mode confined throughout the length of the antenna so that the phasefront of the coupled light across the cross-section of the antenna is flat. At one end of the antenna active area, a mode converter comprising a taper couples the light from 0.82 μm waveguides to 0.5 μm waveguides. A Y-junction-based 16-to-1 combiner tree combines all the outputs from a single antenna into a single mode propagating in the 0.5 μm wide waveguide that is used to route the signal on the PIC.

Three grating regions with apodized coupling strengths are designed (see Section IA in the Supplementary Information), as seen in Fig. 2. The physical footprint of the antenna is 597 × 16.7 μm². Across the length of the antenna, the splitter tree region is from 0 μm to 47 μm, the apodized grating duty cycle region is from 47 μm to 347 μm, the apodized grating width region is from 347 μm to 547 μm, and the full width region is from 547 μm to 597 μm. The aperture of the chip comprises 32 of these antennas with 17.5 μm pitch to ensure sufficiently low optical crosstalk between the antennas. Two antennas are added on each side of the aperture, resulting in 36 total antennas. On each side, one antenna is connected to a standard grating coupler and the other is connected to a photodiode to aid free-space alignment with an optical measurement and an electronic measurement, respectively. By measuring the power at these antennas, the PIC aperture can be aligned more easily to the incident beam. Furthermore, these extra antennas ensure that the edge antennas of the aperture have the same response as the middle antennas.

The QRX design comprises a tunable Mach-Zehnder interferometer (MZI) made out of two 50:50 directional couplers and two diode phase shifters. Each phase shifter is 100 μm long, comprising a resistive heater made out of doped silicon with 1 kΩ resistance and a diode in series with 1 V forward voltage (see Section IB Supplementary Information). Doped Si is placed 0.9 μm away from the waveguides to minimize loss from free carriers. The MZI is configured in a push-pull configuration to extend the tuning range of the coupling coefficients and is designed to provide sufficient tuning with ±5 V drivers. One branch of the MZI includes an optical delay with 90° phase shift to set the nominal coupling of the MZI to 50:50. Fabrication imperfections such as changes in the gap of the coupling region and surface roughness on the waveguides shift the ideal 50:50 coupling randomly across different channels. The tunability of the MZIs allows for the correction of these imperfections to set 50:50 coupling. The MZIs are also designed to be symmetric to ensure a high extinction ratio.

After the MZI, the waveguides are adiabatically tapered to connect to a balanced Ge photodiode pair with >20 GHz bandwidth at 3V reverse bias, >70% quantum efficiency, and <100 nA dark current. The

QRX is surrounded by a Ge shield to absorb stray light propagating in the chip substrate and prevent it from coupling to the photodiodes. Each QRX output is connected to a separate on-chip pad to be interfaced with a transimpedance amplifier and subsequent electronics for RF processing.

The LO is coupled to the chip with a standard grating coupler and is sent to each QRX through a 1-to-32 splitter tree. Each Y-junction in the splitter tree has 0.28 dB loss, and the grating coupler has 3.30 dB loss. Before the splitter tree, a directional coupler on the LO waveguide couples 1% of the LO power to a monitor photodiode for LO power monitoring. After the splitter tree, a TOPS is included in each branch to tune the LO phase of each channel for phase calibration. Each TOPS for phase tuning is 315 μm long, comprising a resistive heater made out of titanium nitride above the waveguide with 630 $\Omega$ resistance.

Further information on the chip component designs and characterization results is given in Section I of Supplementary Information.

## Chip losses

Expected on-chip losses consist of 3.78 dB from simulated antenna insertion loss, 0.321 dB from waveguide propagation loss, and 1.52 dB from photodiode quantum efficiency. This results in a total expected on-chip loss of 5.62 dB. The on-chip losses are verified experimentally by sending 200 μm collimated beam to the chip aperture after setting all QRXs to the unbalanced (100:0) configuration and summing all QRX currents. For 0.452 mW input power, the output current is 0.0615 μA, resulting in an insertion loss of 8.66 dB. In this measurement, in addition to on-chip losses, there is also the geometric loss due to the mode mismatch between the aperture and the collimated beam as well as the insertion loss of the collimator. For a 200 μm collimated beam, the geometric loss is 1.14 dB, the insertion loss of the collimator is 0.8 dB, and the insertion loss of the connectors is expected to be <1 dB. De-embedding these losses from the measurement, the on-chip losses are measured to be 5.72 dB, which agrees well with the 5.62 dB expected loss. Other losses for all of the measurement setups are outlined in Section VIII of Supplementary Information.

## Squeezed light generation

To generate squeezed light, continuous wave light from a fiber-coupled 1550 nm laser is split into a signal path and an LO path. The light in each path is amplified by an erbium-doped fiber amplifier. After amplification in the signal path, the 1550 nm coherent light is upconverted to 775 nm by a PPLN waveguide via second harmonic generation (SHG). The upconverted light is used as a continuous-wave pump for Type 0 spontaneous parametric downconversion (SPDC) with another PPLN waveguide, which generates broadband light in a squeezed vacuum state at a central wavelength of 1550 nm. The characterization of the PPLN waveguide used for SPDC in each experiment is in Section IV of Supplementary Information. The squeezed light is sent to a fiber-optic collimator, which transmits the light over free space with a uniform phase front to the chip aperture. After amplification in the LO path, the 1550 nm coherent light is sent to a bulk lithium niobate electro-optic modulator for phase control. The phase-modulated LO is sent to a cleaved fiber, which is coupled to the LO input of the chip. Polarization controllers before the collimator and on the LO fiber are used to optimize the coupling efficiency to the chip.

## System electronics

The PIC is packaged with an interposer board for fanning the 104 electronic input/output (IO) to/from the chip. The interposer board is designed with a laser-milled cavity in the middle to place the PIC surrounded by pads with blind vias for high-density routing. The chip and the interposer are assembled so that the on-chip pads are level and parallel with the on-board pads to shorten the bond wire length. The traces from the interposer pads to the TIA inputs on the motherboard are minimized and spaced apart sufficiently to minimize electronic

crosstalk with 50 $\Omega$ coplanar waveguide (CPW) transmission lines. The TIA circuit on the motherboard utilizes a FET-input operational amplifier (op-amp) with resistive feedback. The op-amp IC (LTC6269-10) has a 4 GHz gain-bandwidth product and is used with a 50 k$\Omega$ feedback resistor. The capacitance of the feedback trace is used to ensure sufficient phase margin while keeping the closed-loop gain greater than 10 since the op-amp is decompensated. A 50 $\Omega$ resistor is placed in series with the output of the TIA for impedance matching and to dampen any oscillations from capacitive loading at the output. The TIA outputs are routed with 50 $\Omega$ CPW transmission lines to a high-speed, high-density connector to route the signals to data acquisition.

The DC voltage across the TIA feedback resistor is used as the error signal for the CMRR correction and drives an integrator circuit with a chopper-stabilized op-amp IC (OPA2187) for low voltage offset, flicker noise, and offset drift. The integrator's unity-gain bandwidth is set close to DC (23 Hz) to dampen any oscillations in the CMRR auto-correction feedback. The integrator's output is fed back to the MZI on the PIC to correct the CMRR continuously. The polarity of the integrator is designed to match the polarity of the push-pull MZI so that the correction circuit always maximizes the CMRR, whether the error signal is a negative or a positive DC signal. The correction is limited by the dark current of each QRX and the offset voltage at the input of each integrator, but offset correction can be applied to each integrator to further maximize the CMRR. A high-speed coaxial cable assembly is used to connect to the motherboard. The cable coming out of the motherboard first connects to a power board, which powers the active electronics on the motherboard. This board also routes the output from two photodiodes, which are connected to the two edge antennas of the aperture, and the output from the monitor photodiode, which is connected to the LO coupler, to current meters for continuous monitoring of the signal and LO alignment on the chip. Another cable then connects the remaining IO to a splitter board that splits the 32 QRX outputs for simultaneous imaging and RF data acquisition. The remaining control lines for tuning the on-chip TOPS are connected to 32 digital-to-analog converters (DACs) for independent phase tuning of each QRX.

## Data acquisition

The 32 QRX outputs after the splitter is connected to boards that host SMA connectors to interface with data acquisition equipment. One board, used for parallelized 32-channel readout, connects to 32 channels of digitizers with 100 MHz bandwidth, 100 MSa/s adjustable sampling rate, and 14-bit resolution. The digitizers are used in high-impedance mode to read out the voltage of each QRX output for squeezed light imaging and during RF measurements. For squeezed light imaging in Fig. 4c, the digitizers are configured to have a sampling rate of 20 MSa/s. The other board, used for RF single channel readout, connects to a 32-to-1 RF power combiner assembly with an operating frequency range of 0.1-200 MHz. The output from the power combiner is connected to the RF signal analyzer (ESA). For squeezed light measurements in Fig. 5b, c, e, f, the ESA is configured to be used in the zero-span mode at a center frequency of 5.5 MHz, with a resolution bandwidth of 2 MHz and a video bandwidth of 5 Hz. Center frequency and resolution bandwidth are selected to maximize the shot noise clearance after a parameter sweep.

## Beamforming

**Phase calibration.** For each angle of incidence, we calibrate the settings for the 32 LO TOPS such that the quadratures for all pixel modes are aligned to the same phase. Precise phase calibration is crucial to prevent additional loss due to vacuum noise leaking into the combined output. Phase calibration is performed with a 1550 nm coherent state transmitted by the collimator, and a 5 MHz phase ramp is applied to the LO before coupling to the chip. The 5 MHz downconverted RF signal after channel combination is used as feedback to the computer

to tune the on-chip TOPS iteratively. Various signal processing schemes and algorithms have been developed for beamforming in classical phased arrays, such as random search, gradient search, direct matrix inversion, and recursive algorithms[54]. We employ a modified gradient search algorithm by sweeping phase settings of on-chip TOPS with an orthogonal mask set. Further information for the phase calibration algorithm is presented in Section VI of Supplementary Information.

**Channel combination.** For the beamforming measurements in Fig. 5, the array mode function for $N$ channels connected to the power combiner is,

$$\mathcal{A}_N(\rho) \approx \sum_{j \in S_N} \frac{1}{\sqrt{N}} e^{-i\phi_j} \mathcal{E}_j(\rho), \tag{9}$$

where $j$ is summed over the set of connected channels ($S_N$). For beamforming, the phase calibration algorithm optimizes the phases, $\{\phi_j\}$, such that all channels are in phase. Starting with only channel 16 connected to the power combiner, the number of connected channels ($N$) is increased by adding channels to the power combiner symmetrically about the center of the array (i.e. one channel ($N = 1$): $S_1 = \{16\}$; two channels ($N = 2$): $S_2 = \{16, 17\}$; three channels ($N = 4$): $S_3 = \{15, 16, 17\}$; etc.). Figure 5b shows the noise power levels as a function of the number of connected channels, $N = 1, \cdots, 32$. In practice, imperfections or parasitic effects such as impedance mismatch or crosstalk affect the array mode function in Eq. (9), which could result in different amplitude weights for different combinations of channels. To fully account for these effects, the mode matching is classically characterized by measuring the signal-to-noise ratio (SNR) of coherent light (see Section VIIB in the Supplementary Information), which is proportional to the array geometric efficiency, $\eta_g \approx |\int \mathcal{A}_N(\rho) \hat{a}_{\text{in}}(\rho) d\rho|^2$, for each channel combination.

**Squeezing level estimation**

The squeezing and antisqueezing levels relative to the shot noise level are estimated from a statistical analysis of the quadrature sample variances or noise powers. For the squeezed light experiments in Fig. 4 (3, 5), quadrature sample variances (noise powers) are acquired for squeezed vacuum and vacuum states over an approximately uniform distribution of phases, and histograms are constructed for the acquired data. The squeezing and antisqueezing levels are estimated from the inflection points of the probability density functions (PDFs) of quadrature variances, which are obtained from the Gaussian kernel density estimates (KDEs) of the histograms. The squeezing and antisqueezing level estimates correspond to the locations of the peak slopes at the left (right) edges of the PDF, respectively. In particular, the quadrature variances for the squeezing and antisqueezing levels are identified from the peaks in the derivative of the KDEs, which provide a well-defined measure of the edges of the quadrature variance distribution. The same estimation procedure applied to the vacuum data yields the standard deviation in the vacuum sample variance (shot noise level). Error bars are obtained from the propagation of the vacuum standard deviation on the squeezing and antisqueezing level estimates. Further information on the estimation procedure is presented in Section III of the Supplementary Information.

**Wigner function calculation**

For the calculations of the Wigner functions in Fig. 4b, the experimental squeezing parameter $r = 1.95$, which corresponds to 16.9 dB generated squeezing, is plugged into the Wigner function $W(Q, P, r, \phi, \eta)$ of a squeezed vacuum state, setting $\phi = 0$ and $\eta = 1$ to obtain the Wigner function at the source. The Wigner function for each pixel mode is obtained by plugging its squeezing parameter ($r = 1.95$),

phase and geometric efficiency into the Wigner function. The phases are estimated from a sinusoidal fit to the quadrature sample variances of each channel over data with approximately uniform phase variation. From the squeezing parameter, the effective efficiency of each channel is estimated using,

$$\eta_j = \frac{(A_j - 1)\exp(2r)}{(\exp(2r) - 1)(A_j + \exp(2r))}, \tag{10}$$

where $A_j = \Delta Q_{j,+}^2 / \Delta Q_{j,-}^2$ is the ratio of the antisqueezing level ($\Delta Q_{j,+}^2$) to the squeezing level ($\Delta Q_{j,-}^2$) of the $j$th pixel. The geometric efficiencies of the channels are calculated from the effective efficiencies of the channels divided by their total sum. The characterization of the squeezing parameter, phases, and geometric efficiencies are in Section VIIA of Supplementary Information.

**Measurement characterization**

For each non-classical measurement in the reported experiments, a classical measurement is also taken to characterize the system. The classical measurements are taken using the same photonic and electronic hardware chain as the non-classical measurements to ensure consistency. For the squeezed light imaging experiment in Fig. 4b–d, a classical multi-pixel image is taken by sending a coherent state as signal while the LO phase is ramped at 5 MHz. The 5 MHz tone from each channel is digitized by the imaging readout, and its corresponding amplitude is measured. For the experiments in Fig. 5b, e, f, a coherent state is sent as the signal while the LO phase is ramped at 5 MHz. The 5 MHz tone at the output of the power combiner is measured on the ESA for each measurement setting. For each channel combination in Fig. 5b, an SNR is calculated by taking the ratio of the signal power to the corresponding shot noise acquired from the squeezed light measurement. Classical data and more detailed analysis are presented in Section VII of Supplementary Information.

**Theoretical modeling.** The theoretical models in Fig. 5 are constructed from classical characterizations of the effective efficiency for each experimental configuration using,

$$\Delta Q_{\pm}^2 = \eta e^{\pm 2r} + 1 - \eta, \tag{11}$$

where $\Delta Q_{\pm}^2$ are the squeezing ($-$) and antisqueezing ($+$) levels relative to the shot noise level, $r$ is the squeezing parameter, and $\eta$ is the effective efficiency of the system.

For Fig. 5b, the model is obtained from Eq. (11) with $\eta \propto$ SNR for each combination of channels. A least-squares fit is performed by taking the proportionality constant ($\eta_c$) to the classical SNR data as the only free parameter, with the squeezing parameter bounded in the range $r = 0.748 \pm 0.019$ (see Section VIIB of the Supplementary Information). Using SNR data normalized to its peak value, we obtain optimal parameters of $\eta_c = 0.021$ and $r = 0.761$ (6.61 dB generated squeezing).

For Fig. 5c, the model is obtained from Eq. (11) with $r = \mu\sqrt{P}$, where $P$ is the SPDC pump power and $\mu$ is the SPDC efficiency, and a least-squares fit is performed taking the $\mu$ and $\eta$ as free parameters. The optimal parameters are reported in the main text, $\eta = 0.016$ and $\mu = 0.038$ [mW]$^{-1/2}$, which matches the the SPDC efficiency of the PPLN waveguide characterized in Section IV of the Supplementary Information and reported in Table II.

For Fig. 5e, the models are obtained from Eq. (11) and $\eta$ proportional to classical beamwidth data for 8 and 32 channels combined. For each data set, a least-squares fit is performed taking the proportionality constant ($\eta_c^{(N)}$) to the classical beamwidth data as the only free parameter, with the squeezing parameter bounded in the range $r = 0.607 \pm 0.015$ (see Section VIIC of the Supplementary Information). Using beamwidth data normalized to their peak powers, we obtain

optimal parameters of $\eta_c^{(8)} = 0.019$, $\eta_c^{(32)} = 0.014$, and $r = 0.611$ (5.31 dB generated squeezing). Using this estimated $r$, the 8 and 32 channel beamwidths are characterized directly from the squeezed light data by extracting the effective efficiencies using Eq. (10). With linear interpolation, angles corresponding to 0.5 effective efficiency are found to calculate the beamwidths.

For Fig. 5f, the models are obtained from Eq. (11) and $\eta$ proportional to the classical radiation pattern of a single antenna. For each data set, a least-squares fit is performed taking the proportionality constant ($\eta_c^{(N)}$) to the classical radiation pattern as the only free parameter, with the squeezing parameter bounded in the range $r = 0.865 \pm 0.043$ (see Section VIID in the Supplementary Information). Using the radiation pattern data normalized to its peak power, we find optimal parameters of $\eta_c^{(8)} = 0.017$, $\eta_c^{(32)} = 0.015$, and $r = 0.908$ (7.89 dB generated squeezing). Using this estimated $r$, the 8 and 32 channel FoVs are characterized directly from the squeezed light data in the same way as beamwidth characterization using Eq. (10).

The squeezing parameters for the models are obtained from independent characterizations of the sources (see Section IV of Supplementary Information). We note that for the measurements in Fig. 5b, c, e, phase calibration was performed once before acquisition of all the data, whereas for the FoV data in Fig. 5f, separate phase calibration was performed for each data point. Imperfect phase calibration contributes to RF loss due to imperfect destructive interference of vacuum terms in the pixel quadratures. Therefore, depending on the phase calibration, different angles exhibited different amounts of RF loss, specifically $\theta = -1°$, which affected the fit in 5f. Loss due to phase calibration can be further minimized with more sophisticated phase calibration algorithms[54]. Further details of the theoretical modeling are in Section VII of Supplementary Information.

## Data availability
Data from this work are available upon request.

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

## Acknowledgements

We thank Pablo Backer Peral for aiding the development of the digitizer script, Esme Knabe for aiding the initial single-channel QRX measurements, Aroutin Khachaturian for technical discussions on the PIC design, Debjit Sarkar for aiding the setup for the high bandwidth QRX measurement and Andrew Mueller for aiding figure design. Support for this work was provided in part by the Carver Mead New Adventures Fund and in part by the Alliance for Quantum Technologies' (AQT) Intelligent Quantum Networks and Technologies (INQNET) program. S.I.D. is in part supported by the Brinson Foundation. M.S. is in part supported by the Department of Energy under Grant No. SC0019219.

## Author contributions

V.G. conceived and led the project, conceived the ideas, including quantum phased arrays, designed the experiments, built the experimental setups, conducted the measurements, performed the data analysis and led the manuscript writing. S.I.D. contributed to the idea conception, experimental design, experimental setups, measurements and data analysis. V.G. developed the chip architecture, the antenna design and the coherent receiver design. A.H. contributed to the chip architecture and the antenna design. V.G. designed, simulated and laid out the photonic chip and the electronics. V.G. and S.I.D. developed the theory and analytical methods. V.G. developed the loss and scaling analysis. R.V. and N.S. contributed to the experimental design, the experimental troubleshooting, the scaling analysis and supported the development of the theory. A.H. and M.S. supervised the project. All authors participated in writing the manuscript.

## Competing interests

V.G., S.I.D., A.H. and M.S. have filed a patent application (Quantum phased arrays, US patent application US 63/457,727; April 6, 2023). The remaining authors declare no competing interests.
