## [Transparent Peer Review file · Nature Communications]

An on-chip phased array for non-classical light

Corresponding Author: Mr Volkan Gurses

Version 0:

Reviewer comments:

Reviewer #2

(Remarks to the Author)

Review of "An on-chip phased array system for non-classical light"

[Summary of article] : This article demonstrates an on-chip phased array system capable of detecting non-classical light, specifically squeezed light, over free space using an on-chip optoelectronic platform.

[General]: Overall, the paper is well-written and technically rigorous. There is a major improvement from the previous version of the manuscript in terms of clarity. The authors have satisfactorily addressed the remarks raised in the previous round of review and I therefore recommend the manuscript for publication.

[Detailed comments]:

Title: Please consider using "detector" or "receiver" instead of "system" in the title. While it's understood that the demonstrated device can also operate as a transmitter, the manuscript focuses on its operation as a receiver. "System" is still ambiguous and beyond the scope of what has been demonstrated.

Figure 1:

Inset "Wigner function" on subfigure c: For the non-expert reader, it may be helpful to add the Wigner function of (unsqueezed) vacuum for comparison.

Line 127: "This active area is large enough for low-loss free-space coupling to the chip over meter-scale distances with off-the-shelf fiber collimators". Could you either provide a justification of this statement, or refer to a derivation of this "meter-scale" order of magnitude?

Line 140: "This is at least an order of magnitude lower than those of the previously reported on-chip aperture designs". This is a significant statement that requires more detail. Please add some sentences on how this improvement was achieved or refer to the corresponding Supplementary Information section where this is explained.

Line 174: "BW_{3dB} is 10.4 MHz". I cannot find a plot of the transfer function that clearly shows that the 3 dB bandwidth is indeed 10.4 MHz. I am looking for a plot that is similar to the one in the high bandwidth configuration S12.d), but for the high-shot noise clearance configuration. Could you add this plot to fig S11?

Figure 5: Thank you for your answers to the remarks from the previous review round and the updates to the manuscript to further clarify the interpretation of the plots. I understand now that to make these plots, only the phase ϕ_j has been tuned, while the gains g_j are kept constant and equal, as confirmed by equation (9). This explains why the anti-squeezing level in fig 5.b) drops for more than 8 combined channels. However, for a full optimization of beamforming, both ϕ_j and g_j should be tuned. Perhaps you could add a half-sentence on why g_j has not been tuned as well.

Reviewer #4

(Remarks to the Author)

Reviewer #2 (Remarks to the Author):

Review of "An on-chip phased array system for non-classical light"

[Summary of article] : This article demonstrates an on-chip phased array system capable of detecting non-classical light, specifically squeezed light, over free space using an on-chip optoelectronic platform.

[General]: Overall, the paper is well-written and technically rigorous. There is a major improvement from the previous version of the manuscript in terms of clarity. The authors have satisfactorily addressed the remarks raised in the previous round of review and I therefore recommend the manuscript for publication.

Thank you for the summary and for your assessment of the work.

[Detailed comments]:

Title: Please consider using "detector" or "receiver" instead of "system" in the title. While it's understood that the demonstrated device can also operate as a transmitter, the manuscript focuses on its operation as a receiver. "System" is still ambiguous and beyond the scope of what has been demonstrated.

Thank you for the recommendation. We have removed "system" from the title to change it to "An on-chip phased array for non-classical light". We believe this is an apt description of the work since the work introduces a novel type of phased arrays.

Figure 1:

Inset "Wigner function" on subfigure c: For the non-expert reader, it may be helpful to add the Wigner function of (unsqueezed) vacuum for comparison.

Thank you for this recommendation. We have added the vacuum Wigner function to Figure 1.

Line 127: "This active area is large enough for low-loss free-space coupling to the chip over meter-scale distances with off-the-shelf fiber collimators". Could you either provide a justification of this statement, or refer to a derivation of this "meter-scale" order of magnitude?

Thank you, we have added references to the corresponding Supplementary Information sections where this is explained.

Line 140: "This is at least an order of magnitude lower than those of the previously reported on-chip aperture designs". This is a significant statement that requires more detail. Please add some sentences on how this improvement was achieved or refer to the corresponding Supplementary Information section where this is explained.

Thank you, we have added references to the corresponding Supplementary Information sections where this is explained.

Line 174: "BW_{3dB} is 10.4 MHz". I cannot find a plot of the transfer function that clearly shows that the 3 dB bandwidth is indeed 10.4 MHz. I am looking for a plot that is similar to the one in the high bandwidth configuration S12.d), but for the high-shot noise clearance configuration. Could you add this plot to fig S11?

3-dB bandwidth of the high-shot noise clearance configuration was measured with only two data points (DC and frequency at which the gain drops by 3 dB). Therefore, we didn't include a frequency response plot for this configuration.

Figure 5: Thank you for your answers to the remarks from the previous review round and the updates to the manuscript to further clarify the interpretation of the plots. I understand now that to make these plots, only the phase ϕ_j has been tuned, while the gains g_j are kept constant and equal, as confirmed by equation (9). This explains why the anti-squeezing level in fig 5.b) drops for more than 8 combined channels. However, for a full optimization of beamforming, both ϕ_j and g_j should be tuned. Perhaps you could add a half-sentence on why g_j has not been tuned as well.

Thank you for this point and recommendation. We indeed didn't have continuous amplitude control on the channels, which could be achieved by tuning the shot noise clearances either with LO power or electronic noise floor. We added this information to lines 275-279.

Reviewer #4 (Remarks to the Author):

Thank you.